# CFD-Based Flow Channel Optimization and Performance Prediction for a Conical Axial Maglev Blood Pump

**DOI:** 10.3390/s22041642

**Published:** 2022-02-19

**Authors:** Weibo Yang, Sijie Peng, Weihu Xiao, Yefa Hu, Huachun Wu, Ming Li

**Affiliations:** 1School of Mechanical and Electronic Engineering, Wuhan University of Technology, Wuhan 430070, China; yangwb@whut.edu.cn (W.Y.); 305000@whut.edu.cn (S.P.); 304933@whut.edu.cn (W.X.); huyefa@whut.edu.cn (Y.H.); tuxing@whut.edu.cn (M.L.); 2Hubei Provincial Engineering Technology Research Center for Magnetic Suspension, Wuhan 430070, China

**Keywords:** blood pump, magnetic bearing, computational fluid dynamics, flow characteristics, advanced turbulence model

## Abstract

Ventricular assist devices or total artificial hearts can be used to save patients with heart failure when there are no donors available for heart transplantation. Blood pumps are integral parts of such devices, but traditional axial flow blood pumps have several shortcomings. In particular, they cause hemolysis and thrombosis due to the mechanical contact and wear of the bearings, and they cause blood stagnation due to the separation of the front and rear guide wheel hubs and the impeller hub. By contrast, the implantable axial flow, maglev blood pump has the characteristics of no mechanical contact, no lubrication, low temperature rise, low hemolysis, and less thrombosis. Extensive studies of axial flow, maglev blood pumps have shown that these pumps can function in laminar flow, transitional flow, and turbulent flow, and the working state and performance of such pumps are determined by their support mechanisms and flow channel. Computational fluid dynamics (CFD) is an effective tool for understanding the physical and mechanical characteristics of the blood pump by accurately and effectively revealing the internal flow field, pressure–flow curve, and shear force distribution of the blood pump. In this study, magnetic levitation supports were used to reduce damages to the blood and increase the service life of the blood pump, and a conical impeller hub was used to reduce the speed, volume, and power consumption of the blood pump, thereby facilitating implantation. CFD numerical simulation was then carried out to optimize the structural parameters of the conical axial maglev blood pump, predict the hemolysis performance of the blood pump, and match the flow channel and impeller structure. An extracorporeal circulation simulation platform was designed to test whether the hydraulic characteristics of the blood pump met the physiological requirements. The results showed that the total pressure distribution in the blood pump was reasonable after optimization, with a uniform pressure gradient, and the hemolysis performance was improved.

## 1. Introduction

Heart failure due to various cardiovascular diseases has become one of the major causes of death in the world. In clinical practice, heart failure is usually treated with heart transplantation, ventricular assist devices, or total artificial heart [1,2]. Heart transplantation can be limited by a shortage of donors, so ventricular assist devices and total artificial hearts have gradually been developed from temporary to permanent implants. Accordingly, the blood pumps used in these implants have seen extensive development, including the utilization of magnetic bearings. Magnetic bearings eliminate the mechanical contact between the bearing and the impeller, thereby achieving a streamlined rotor structure and minimizing hemolysis and thrombosis. Hence, implantable maglev blood pumps are superior to traditional blood pumps [3,4,5]. Computational fluid dynamics (CFD) is a powerful tool for improving the design of blood pumps, as it can accurately and effectively reveal the internal flow field, pressure–flow curve, and internal shear force distribution of the blood pump. Moreover, the splitter blade can effectively optimize pump performance. It is very important to research these features [6,7]. Combining CFD with experimental verification is critical for developing a deeper understanding of the blood flow characteristics of blood pumps.

The blood flow channel and impeller parameters directly affect the performance of the blood pump and determine the performance of blood circulation. Improper design of the impeller and flow channel can cause hemolysis and thrombosis in the blood pump. Moreover, there is complicated non-linear turbulent flow in the blood pump, and it remains a major challenge to accurately obtain the turbulent flow model. Reynolds-averaged Navier–Stokes (RANS) models are currently the most widely used in scientific research and engineering. These are mainly two-equation turbulence models, such as the standard κ-ε turbulence model, realizable κ-ε turbulence model, standard κ-ω turbulence model, and shear stress transfer (SST) κ-ω turbulence model. In κ-ε models, *k* represents turbulent pulsation kinetic energy, whose unit is the joule, *ε* represents dissipation rate of turbulent pulsation kinetic energy. The larger the *k*, the larger the turbulent pulsation length and the time scale, whereas the larger the *ε*, the smaller the turbulent pulsation length and the time scale. They are two quantities that restrict the turbulent pulsation. In κ-ω models, *ω* means the dissipation rate of energy per unit volume and unit time. Since the actual dissipation process occurs in the smallest scale vortex, the turbulent kinetic energy dissipated per unit time is the turbulent kinetic energy transferred to the smallest scale vortex per unit time. The characteristics and scope of application of the above models are different. Among them, the SST κ-ω turbulence model has the characteristics of the standard κ-ε turbulence model and standard κ-ω turbulence model and thus has wide applicability and good reliability. Hence, the SST κ-ω model has great advantages for calculating the blood flow characteristics of blood pumps.

In blood pump design, CFD is mainly used to simulate hemolysis and thrombosis performance. Liu et al. [8] analyzed the influence of a bearing structure in the axial blood pump on the potential of device thrombosis, and the results showed that the platelets flowing through the bearing area were exposed, leading to the adhesion of activated platelets. Wu et al. [9] used CFD modeling provided valuable information in determining the appropriate parameters of diffuser to improve pump efficiency and avoided excessive fluid shear stress. Yano et al. [10] used the Lagrange particle tracking method to verify the hemolysis model. Taskin et al. [11] built an experimental device to study the difference between the Euler model and the Lagrange power model in hemolysis prediction, which showed that the Lagrange power model was more accurate. Rezaienia et al. [12] studied the effects of axial and radial clearance on the fluid dynamics and hemolysis of a centrifugal blood pump. It was found that the optimal clearance was only suitable for a specific impeller geometry.

In view of the above problems, the present study used an integrated rotor design to reduce the generation of blood stagnation, thereby reducing the risk of thrombosis. A conical design was used for the hub of the impeller, which increased the pressure of the blood pump. While meeting design requirements, the axial blood pump reduced the axial size of the impeller as well as the speed, thereby reducing power consumption and the hemolysis rate and improving the survival rate.

## 2. Materials and Methods

### 2.1. Description of the Conical Axial Maglev Blood Pump

A new conical axial blood pump with active magnetic bearing was designed, which can avoid thrombus caused by contact between rotor and stator.

The conical axial maglev blood pump is a left ventricular auxiliary device supported by electrodynamic bearings in the radial direction and electromagnetic bearings in the axial direction, and it has a conical impeller hub (Figure 1). The front and rear guide wheel blades are fixed inside the blood pump. The front guide wheel eliminates the circumferential speed component of the blood flow entering the blood pump, and the rear guide wheel eliminates the circumferential speed component of the blood flow and converts the kinetic energy into pressure. The impeller and the front and rear hubs are integrated, and the impeller hub is in a cone shape. In order to ensure the bearing capacity of the radial electrodynamic bearing, the taper angle is not too large. The radial electrodynamic bearings control the radial displacement of the rotor, and the axial electromagnetic bearings control the axial displacement of the rotor. Together, these bearings achieve magnetic suspension of the rotor so as to eliminate mechanical contact and reduce hemolysis caused by friction.

The conical axial maglev blood pump is mainly composed of the pump body, impeller, and front and rear guide wheels. The outer diameter of the casing is 30 mm, the axial length is 70 mm, the hub is 11.6 mm in diameter, and the range of rotating speed is 8000–14,000 rpm.

### 2.2. Experimental Setting

The extracorporeal circulation simulation platform included the blood pump, pipelines, and the data collection unit. The platform was used to test the hydraulic parameters of the pump. When the system is working in a stable condition, the flow at any position is basically equal, while the pressure changes. Pressure gauges were installed at both ends of the blood pump, and a flow meter was installed at the outlet of the liquid storage tank. The flow of the blood pump could be adjusted via the damping valve. The platform is shown in Figure 2.

The pressure gauges were MIK-P300, with a DC24V power supply, the range was 0–50 kPa, and the accuracy was ±0.03% FS/°C. The damping valve was IEV2-10.2-TM/R with an adjustable opening. The flowmeter was LWGY-6C, with a DC24V power supply, and the accuracy was ±1%. The pipeline was made of PVC with an inner diameter of 12 mm, and the inner surface was smooth to help reduce peripheral resistance. The liquid storage tank was a glass tank with a volume of 2 L.

### 2.3. Preparation of the 3D Model

A three-dimensional model of the conical axial maglev blood pump was created, and it is displayed in Figure 3. It mainly consisted of the fluid components, supporting components, and the driving component. The fluid components included the blood flow channel, the impeller, and the front and rear guide wheels. The supporting components were composed of radial electromagnetic bearings and axial electrodynamic bearings. The driving component was a permanent magnet brushless DC motor, which drives the impeller to transfer the blood from the front guide wheel to the rear guide wheel area.

### 2.4. Computational Domain

In the SolidWorks software, the cavity function was used to build the flow channel model of the blood pump. According to the actual state of the blood pump, the dynamic and static areas of the blood pump were divided, forming three separate areas with different settings.

### 2.5. Mesh Generation

Meshing plays a decisive role in the success of CFD simulation and simulation accuracy. There are different types of meshes available, among which structured mesh, unstructured mesh, and hybrid mesh are the most comment types. In a structured mesh, the adjacent nodes have a clear sequence and relationship. Boundary nodes can have different numbers of adjacent elements, whereas the internal nodes all have the same number of adjacent elements. Specifically, for 1D, 2D, and 3D nodes, there are two, four, and six adjacent elements. Compared with a structured mesh, the nodes of an unstructured mesh are irregular and disordered, so the number of adjacent elements for each node may be different [13]. In a hybrid mesh, there are both types of meshes, and the nodes at different positions have different connections.

Since the flow channel of the blood pump has a complex structure, and the impeller has a curved surface, it is difficult to use a structured mesh. Thus, the automatic meshing method in the ANSYS Fluent 12.1.2 (ANSYS Inc., Canonsburg, PA, USA) software was used. The model obtained in this method is highly adaptable and suitable for various types of flow channels.

For the conical axial maglev blood pump in this study, the front and rear guide wheels of the blood pump were in non-rotating states, and the impeller was in a rotating state. Therefore, the multiple reference frame (MRF) model was used to set the rotation area. The flow channel model was divided into three parts, which were meshed with non-structured elements [14,15]. The gradient of physical quantities in the rotating area was large, so the meshes were refined. Figure 4 shows the blood pump model, where Figure 4a–d are the meshes of the flow channel, front guide wheel, impeller, and rear guide wheel, respectively.

### 2.6. Material Properties

The boundary conditions were considered to be mass flow at the input and static pressure at the output. The impeller zone was described as a revolving frame, and the wall was specified as a non-slip wall. The calculations were performed for steady-state flow, with constant boundary conditions over time. In each CFD simulation of the blood pump, maximal convergence criteria of 10^−4^ were applied for velocity, turbulent kinetic energy, turbulent dissipation rate, and continuity [16,17]. In addition, the smaller the iteration residual value is set, the more accurate the results of the simulation are. The default value of the iteration residual of fluent software is 10^−3^, and if more accurate results are sought, the iteration residual value should be smaller than the default value, but the smaller the value, the greater the computer memory occupied, leading to a very long calculation process, so when selecting the iteration residual value, both the accuracy of the simulation result and the time to obtain it should be considered, and the balance between the two sides should be taken into account. Moreover, many related studies in the literature also use 10^−4^ as their iterative residual value. Based on the reasons above, in this study, the iterative residual value was chosen as 10^−4^.

The fluid medium of the conical axial maglev blood pump is blood. The blood flow of an adult is on average 4.5–6.8 L/min, and in this study, the blood flow was set to 6 L/min. In order to meet the physiological needs of the human body, the minimum pressure difference between the inlet and outlet of the blood pump was 100 mmHg (i.e., 13.3 kPa). The blood viscosity was set to 0.0035 kg/(m·s), and the density was set to 1050 kg/m^3^ [18].

### 2.7. Boundary and Initial Conditions

According to the structure of the conical axial maglev blood pump, it was divided into three sections: the front guide wheel section, the impeller section, and the rear guide wheel section. The front and rear guide wheel sections were static regions, whereas the impeller section was a rotating region.

Due to the different functions of each part of the blood pump, different boundary conditions need to be set for each contact surface with the blood, and therefore, they are mainly divided into the following sections: inlet, wall (including inducer wall, impeller wall, diffuser wall), interface, and outlet. The Inlet section is the entrance into which the blood flows. The outlet is the exit where the blood flows out. Wall boundary is a no-slip smooth wall. In order to simplify the motion state of different sections, the MRF model was used to set the rotation area and rotating speed. Information was transmitted between the rotating section and the static sections through Interface. The interface is a type of boundary, i.e., the boundary of the computational domain. Therefore, in order to keep the computational domain unblocked in multiple computational domains, it is necessary to create an interface on the boundary that is in contact with each other. The function of the interface is used to solve the problem that the volume meshes on both sides of a certain surface do not share nodes, and the data transmission of nodes is realized by setting the interface. The inlet was set to mass–flow–inlet, which specifies the mass flow rate at the inlet for compressible flow and gives the mass flow rate on the inlet boundary, the local total inlet pressure is changed at this time to adjust the speed to achieve the given flow rate, which makes the convergence speed of the calculation slower. Additionally, the outlet was set to outflow, which means free outlet. This boundary condition is used to simulate that the outlet velocity or pressure cannot be known before the problem is solved; the outlet flow conforms to the fully developed condition, and at the outlet, except for the pressure, the gradients of other parameters are zero. These settings were related to the actual working conditions of the blood pump, especially with the temperature and directions of each part. Since the walls in different regions were all stationary, the walls were set as stationary walls, while the blades of the impeller were rotating, so the blades were set as rotating walls. The boundary conditions of the blood pump are shown in Figure 5.

The Re of the blood pump was 19,110, which is above 3000, thus indicating turbulent flow. Therefore, it was necessary to select a suitable turbulence model during simulation. Among the current turbulence models, the κ-ε model is the most common since it has high accuracy for free shear turbulence, attached boundary layer turbulence, and moderately separated turbulence. Therefore, the standard κ-ε model was selected in this study.

### 2.8. Solver Settings and Turbulence Modeling

Mass equation: The analysis of any fluid flow state needs to satisfy the differential expression of the continuity equation as follows:(1)$$\frac{\partial \rho }{\partial t}+\frac{\partial \left(\rho u_{x}\right)}{\partial x}+\frac{\partial \left(\rho u_{y}\right)}{\partial y}+\frac{\partial \left(\rho u_{z}\right)}{\partial z}=0$$
where $u_{x}$, $u_{y}$, and $u_{z}$ are the speed components (m/s) in the *x*, *y*, and *z* directions, respectively; t is time (s); ρ is density (kg/m^3^). The continuity equation is a general flow continuity equation and should be used in combination with the actual boundary conditions in specific applications. For example, if the flow is a steady flow, the velocity, pressure, and flow at any point in the flow field are independent of time. At this time, the value of the partial derivative of the continuity equation with respect to time is zero. For another example, if the flow is a univariate steady flow, the continuity equation can be further simplified. The univariate flow is only related to the change in one direction and is entirely irrelevant to changes in the other two directions; therefore, assuming that it is only related to the x direction, it will be not be related to *y* and *z* directions. The partial differential of the direction is zero, and since it is a steady flow, the term of the partial differential to time is also zero. Therefore, in a one-variable steady flow, the left side of the continuity equation has only one term related to the partial differential of *x*. It can be seen that the above equations are closely related to the initial boundary conditions of the fluid.

Momentum equation: suppose velocity $\vec{u}=\left(u_{1},u_{2},u_{3}\right)$ and define ∇=(∂/∂x,∂/∂y,∂/∂z) as the meaning of multiplication ∇. The rate of change in the momentum of the system with respect to time is equal to the resultant external force on the system; that is,
(2)$$\frac{\partial \left(\rho u_{x}\right)}{\partial t}+\nabla \cdot \left(\rho u_{x}\vec{u}\right)=\frac{\partial p}{\partial x}+\frac{\partial \tau_{xx}}{\partial x}+\frac{\partial \tau_{y}{}_{x}}{\partial y}+\frac{\partial \tau_{z}{}_{x}}{\partial z}+\rho f_{x}$$
(3)$$\frac{\partial \left(\rho u_{y}\right)}{\partial t}+\nabla \cdot \left(\rho u_{y}\vec{u}\right)=-\frac{\partial p}{\partial y}+\frac{\partial \tau_{xy}}{\partial x}+\frac{\partial \tau_{yy}}{\partial y}+\frac{\partial \tau_{zy}}{\partial z}+\rho f_{y}$$
(4)$$\frac{\partial \left(\rho u_{z}\right)}{\partial t}+\nabla \cdot \left(\rho u_{z}\vec{u}\right)=-\frac{\partial p}{\partial z}+\frac{\partial \tau_{xz}}{\partial x}+\frac{\partial \tau_{y}{}_{z}}{\partial y}+\frac{\partial \tau_{y}{}_{zz}}{\partial z}+\rho f_{z},$$
where *p* is the pressure on the fluid element (Pa); $\tau_{x}{}_{x}$, $\tau_{xy}$, and $\tau_{xz}$ are the components of viscous stress τ (Pa); $f_{x}$, $f_{y}$, and $f_{z}$ are the unit mass forces (m/s^2^) in the *x*, *y*, and *z* directions, respectively. If the only external force is gravity, then $f_{x}=f_{y}=0$, $f_{z}=-g$. Similar to the continuity equation above, the momentum equation for the fluid here applies to any three-dimensional unsteady flow, compressible or incompressible, viscous or inviscid. Indeed, similar to the general continuity equation, the specific use of the fluid momentum equation is also closely related to the initial boundary conditions of the fluid. For example, if the flow is a steady inviscid flow, then the dynamic viscosity u in this equation is zero, and the fluid. The rate of change in the physical quantity with time is zero, and the equation is simplified—the famous Euler equation.

Energy equation: The energy equation is the law of conservation of energy—that is, the first law of thermodynamics,
(5)$$\frac{\partial \left(\rho E\right)}{\partial t}+\nabla \cdot \left[\vec{u}\left(\rho E+p\right)\right]=\nabla \cdot \left[k_{eff}\nabla T-\sum_{j}h_{j}J_{j}+\left(\tau_{eff}\cdot \vec{u}\right)\right]+S_{h},$$
where E is the total energy of fluid micelles, $E=h-p/\rho +u^{2}/2$ (J/kg); *h* is the enthalpy of the fluid micelles.$h_{j}$ is the enthalpy of component j (J/kg); $k_{eff}$ is the effective thermal conductivity, $k_{eff}=k+k_{t}$ (W/(m∙K)); *k* is thermal conductivity; $k_{t}$ is the turbulent conduction coefficient; $J_{j}$ is the diffusion flux of component j; $S_{h}$ is the volume heat source term.

Equation (5) is the first law of thermodynamics, which is related to the law of energy conservation and transformation in the field of thermal phenomena, reflecting that different forms of energy are conserved in the process of transfer and transformation. According to the second law of thermodynamics, heat cannot spontaneously transfer from a low-temperature object to a high-temperature object. This theorem specifies the direction of energy exchange between objects. The effective thermal conductivity will be influenced by the material characteristics of the boundaries.

Actually, blood is a non-Newtonian fluid, that is, the gradient of viscous shear stress size and velocity is not a purely linear relationship but depends on the relevant references—in a state of high shear rate, blood exhibits the properties of a Newtonian fluid, and in this state, it can be considered a Newtonian fluid for analysis. Therefore, this article analyzed blood as a Newtonian viscous fluid.

### 2.9. Mesh Independence Test

The impeller section of the blood pump was a rotating area with a large gradient of physical quantities. Thus, mesh refinement was carried out by modifying the element size, interface element accuracy, and area of refinement. Mesh independence analysis was carried out by varying the number of elements from 120,000 to 4.67 million. With the same blood pump parameters and boundary conditions, the outlet speeds were compared (Figure 6). It can be seen that when the number of elements was above 3.2 million, the outlet pressure changed only slightly. Thus, the number of elements was set to 3.2 million for the following simulations.

## 3. Flow Channel Optimization and Analysis

### 3.1. Optimization of Front and Rear Tips of Rotor Hubs

First, the design principles of the traditional axial blood pump and the arc method were employed to design the impeller structure. The tip of the front guide wheel hub of the blood pump can reduce the resistance to blood flow, thereby increasing the speed of blood at the inlet and shortening the blood flow time in the inlet area. The tip of the rear guide wheel hub can reduce the probability of blood flow separation and reduce vortex resistance. The two tips play an important role in stabilizing the internal flow field of the blood pump.

The hub tip structure obtained by streamlined design was denoted as design Figure 7a, and the structure obtained by manual adjustment was denoted as design Figure 7b. The simulation result of the speed vector of the hub tip of the rear guide wheel is shown in Figure 7.

It can be seen from Figure 7 that when blood flowed through the hub of design Figure 7a, the circumferential speed component was small, and the vortex was small and scattered and had little effect on the flow field. In design Figure 7b, the circumferential speed component was large, and a large and concentrated vortex was generated. The presence of the vortex increased the probability of thrombosis and hemolysis and increased energy loss, thereby reducing the efficiency of the blood pump.

Figure 8 shows the speed vector diagram of the central section of the rear guide wheel. It can be seen that the blood speed distribution in the rear guide wheel area was uniform, without large speed changes. In design Figure 8a, the blood flowed directly to the outlet of the blood pump in the rear guide wheel area without backflow. In design Figure 8b, there was an area of blood stagnation at the tip of the rear guide wheel hub, and backflow occurred at the center of the outlet area, which increased the probability of hemolysis and thrombosis, causing pressure loss of the blood pump and reducing the efficiency of the blood pump. Therefore, to obtain an optimal flow channel structure, the streamlined design of the front and rear hub tips was used in this study.

### 3.2. Integrated Rotor Design

Since magnetic bearings were used to support the rotor of the blood pump, the front and rear guide wheel hubs could be integrated with the impeller hub. Under the premise of the same structural parameters, the non-integrated design Figure 9a and the integrated design Figure 9b were compared. Under the conditions of 6 L/min and 11,000 r/min, the total pressure distribution on the central cross-section is shown in Figure 9, and the streamline diagram is shown in Figure 10.

It can be seen that in the non-integrated design, there was clearance between the front and rear guide wheel hubs and the impeller hub (Figure 9a), and in this clearance, there was a negative pressure zone, which produced secondary flow from the high-pressure zone in the flow channel to the low-pressure zone. The backflow resulted in blood stagnation, which could lead to thrombosis and endanger the life of the patient in severe cases. In comparison, for the blood pump with the integrated rotor design, blood was pressurized by the rotating impeller and flowed to the rear guide wheel. The total pressure distribution in the impeller area increased progressively without abrupt changes. The total pressure distribution in the flow field was even, and the blood stagnation phenomenon was eliminated.

From Figure 10a, it can be seen that the non-integrated design had a clearance between the front and rear guide wheel hubs and the impeller hub. The blood rotated around the clearance, increasing the total travel distance of the blood in the blood pump, which increased the exposure time of blood to shear stress in a non-physiological state, thereby increasing the probability of blood cell fragmentation and the occurrence of hemolysis. By contrast, as shown in Figure 10b, the integrated design had smooth blood flow without backflow, thus reducing the probability of hemolysis.

### 3.3. Conical Optimization of the Impeller Hub

The impeller hub of traditional axial blood pumps is typically cylindrical. In this study, the impeller hub was conical, and the distance from the hub to the edge of the cavity gradually decreased, so the flow area from the inlet to the outlet became increasingly smaller. As a result, the pressure of the blood squeezed by the impeller blades gradually increased, and the outlet pressure of the conical axial maglev blood pump was higher than that of traditional axial blood pumps. The conical design reduced the rotor speed and the overall size of the blood pump while meeting the pressure requirement, which is of great significance for the clinical application of the blood pump. Figure 11a shows the rotor of a traditional axial blood pump, and Figure 11b shows a rotor with a taper angle of 2°.

A CFD simulation was carried out under the conditions of 11,000 r/min and 6 L/min, and it was obtained that the inlet and outlet pressure difference in the traditional axial blood pump was 3919 Pa and that in the conical axial maglev blood pump was 359.2% higher, at 17,995 Pa.

According to the above results, the structural parameters of the conical axial maglev blood pump were determined as follows: inner cavity clearance, 0.3 mm; the number of blades of the rear guide wheel, 5; hub taper angle, 0.75°. The final taper angle needs to be determined based on hemolysis analysis.

## 4. Hemolysis Performance Prediction

In this section, CFD analysis is discussed, which was carried out to analyze the flow field of the optimized axial blood pump. The power function model proposed by Giersiepen et al. [19] was used to establish the hemolysis prediction model, and the particle tracking method was adopted to obtain the flow trajectory of a certain blood cell, thereby predicting the hemolysis performance of the blood pump.

The mathematical equation is as follows:(6)dHbHb[00]=3.62·10−7·τ2.416·t0.785
where *Hb* is the total amount of hemoglobin, *dHb* is the amount of free hemoglobin caused by hemolysis, t is the exposure time (s), and *τ* is the shear stress (N/m^2^). The shear stress consists of two parts: turbulent shear stress (Reynolds stress) and molecular shear stress.

The particle tracking method was used to predict the hemolysis of the blood pump, and the particle trajectory was plotted. The trajectories of 200 particles were randomly selected, and the hemolysis prediction value $D_{p,i}$ for each blood cell was calculated. The results are shown in Figure 12.

The results showed that the hemolysis prediction values of a large number of blood cells were lower than 0.004, and the average hemolysis value was $\bar{E}$ = 0.00232, indicating that the hemolysis performance of the blood pump was excellent.

## 5. Extracorporeal Circulation Experiment

The outlet pressure curve of the conical axial maglev blood pump was obtained for different hub taper angles. In order to facilitate the analysis, the errors between simulation results and the experimental results were plotted in the same graph, as shown in Figure 13.

It can be seen from Figure 13 that the trend of the simulation curve was consistent with that of the experimental results, and the experimental data were always lower than the simulation results, with an error of about 10%. Thus, CFD simulation was able to accurately analyze the influence of the impeller hub taper angle on the hydraulic performance of the blood pump. Moreover, the experimental results showed that as the hub taper angle increased, the outlet pressure of the blood pump constantly increased, which verified the effectiveness of the conical design in improving the hydraulic performance of the blood pump.

The outlet pressure curve of the blood pump was obtained at different rotating speeds. In order to facilitate the analysis, the errors between simulation results and the experimental results were plotted in the same graph, as shown in Figure 14.

It can be seen from Figure 14 that the trend of the simulation curve was consistent with that of the experimental results, and the impeller speed had a large influence on the outlet pressure of the blood pump. The sources of experimental errors included the equivalent design of the rotor support and the pressure loss of the blood in the channel. When the speed was less than 9000 r/min, the pressure was low, and the error caused by the pressure loss led to a large difference between the simulation data and the experimental results. When the speed was greater than or equal to 9000 r/min, the error caused by the pressure loss was relatively small, which was about 15%. When the speed was 11,000 r/min, the outlet pressure was 91 mmHg, which did not meet the physiological requirements of the human body. When the speed increased to 12,000 r/min, the outlet pressure was 110 mmHg, which met the physiological requirements.

In summary, the experimental platform was able to simulate extracorporeal circulation, and the conical axial maglev blood pump with optimized flow channel design performed well in the experiment. Comparing the experimental results and the simulation results indicates that there may be errors, and the main reason for the errors may be the difference in the simulation boundary conditions and mesh generation, as boundary conditions in the experimental process were not completely consistent with the simulation. The blood pump met the physiological requirements of the human body by increasing the blood pump’s rotating speed.

## 6. Conclusions

Based on an optimization analysis of the traditional axial blood pump, the hub tip obtained by the streamlined design method was reasonable, and the speed field in the hub area was uniform, with only a small degree of vortex and backflow. By comparison, the manually adjusted structure had a relatively concentrated vortex and backflow phenomenon. The integrated design of the rotor eliminated the blood stagnation area and reduced the bypass phenomenon, thereby improving the hemolysis performance of the blood pump. Due to the conical design of the impeller hub, the outlet pressure of the blood pump increased greatly, indicating good effectiveness of the optimal design.Based on the hemolysis prediction analysis of the blood pump, the average hemolysis value of the blood pump was E = 0.00232, which was much smaller than that of traditional axial blood pumps. The high shear force had a large influence on the hemolysis prediction value, indicating that in the design process of axial blood pumps, it is necessary to avoid high shear stress areas. The hemolysis prediction value of the blood pump was negatively correlated with the clearance of the blood pump. Thus, under the premise of meeting working conditions, the clearance should be as large as possible.An extracorporeal circulation platform was built to test the performance of the blood pump. As the taper angle of the impeller hub increased, the outlet pressure also increased, which verified the effectiveness of the conical design in improving the hydraulic performance of the blood pump. There were certain differences between the experimental results and the simulation data; nevertheless, the overall trend was consistent, which verified the correctness of the simulation results.

## Figures and Tables

**Figure 1 sensors-22-01642-f001:**
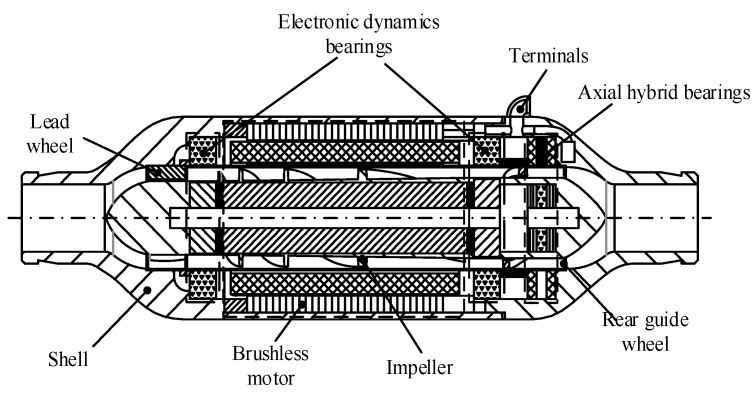
Structure of the conical axial maglev blood pump.

**Figure 2 sensors-22-01642-f002:**
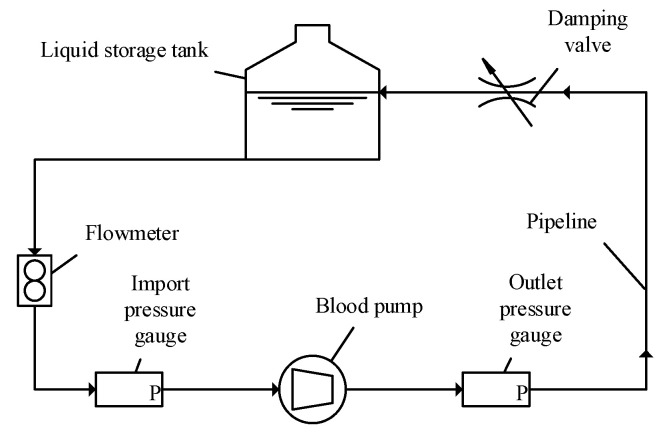
Schematic diagram of the extracorporeal circulation experiment platform.

**Figure 3 sensors-22-01642-f003:**
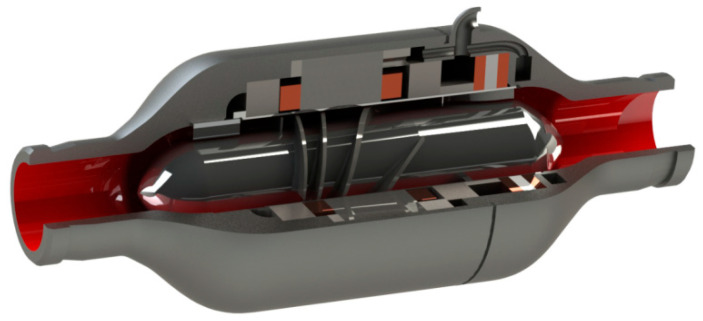
Three-dimensional model of the conical axial maglev blood pump.

**Figure 4 sensors-22-01642-f004:**
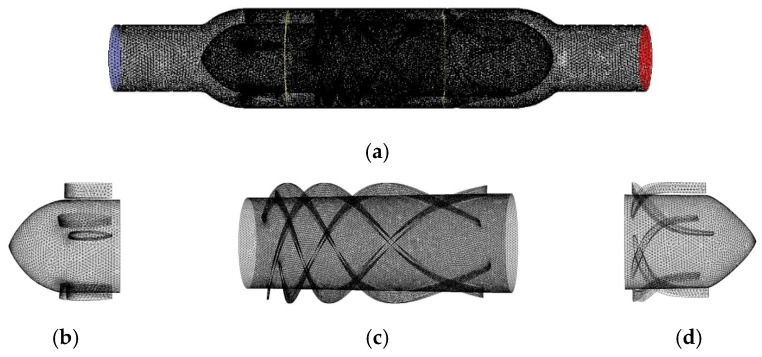
Meshes of the (**a**) whole flow channel, (**b**) front guide wheel, (**c**) impeller, and (**d**) rear guide wheel.

**Figure 5 sensors-22-01642-f005:**
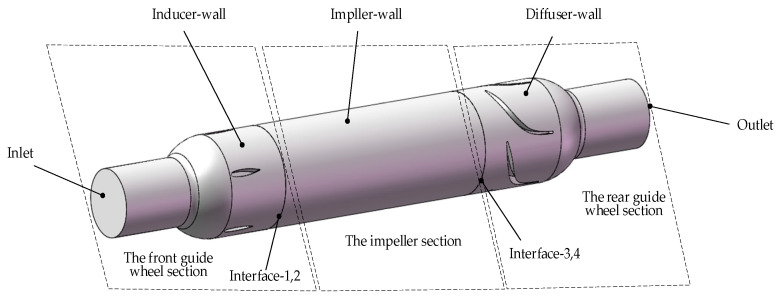
Setting boundary conditions of the blood pump.

**Figure 6 sensors-22-01642-f006:**
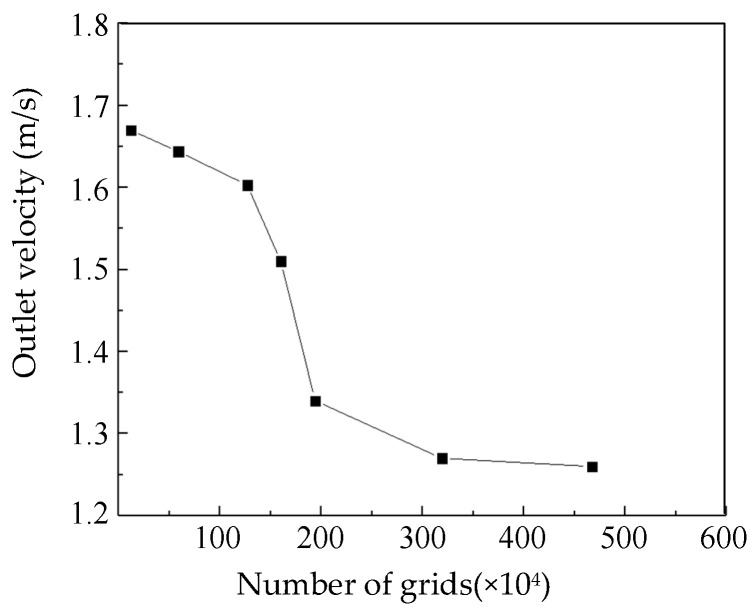
Mesh sensitivity analysis.

**Figure 7 sensors-22-01642-f007:**
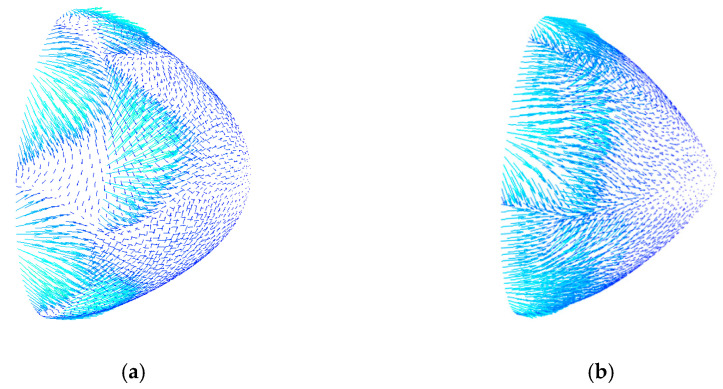
Speed vector distribution at the tip of the hub for the (**a**) streamlined design and (**b**) manual adjustment design.

**Figure 8 sensors-22-01642-f008:**
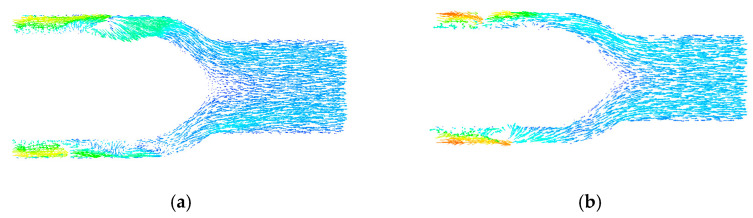
(**a**,**b**) Speed vector distribution of the center section of the rear guide wheel.

**Figure 9 sensors-22-01642-f009:**
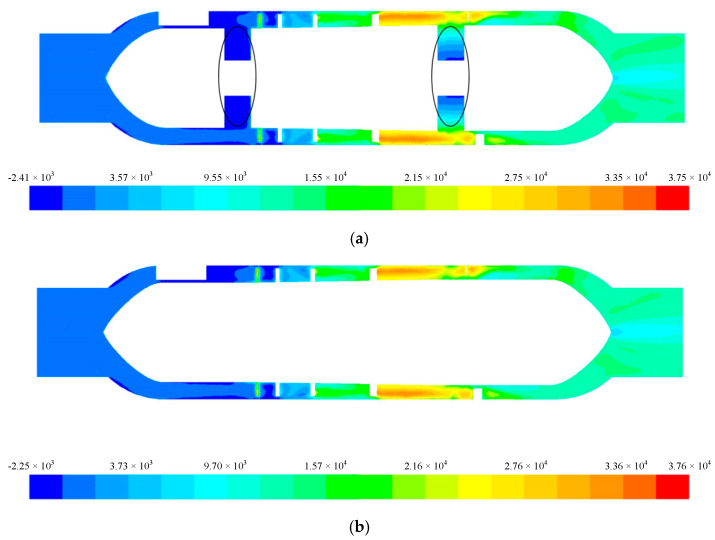
Total pressure distribution of the central section for the (**a**) non-integrated design and (**b**) integrated design.

**Figure 10 sensors-22-01642-f010:**
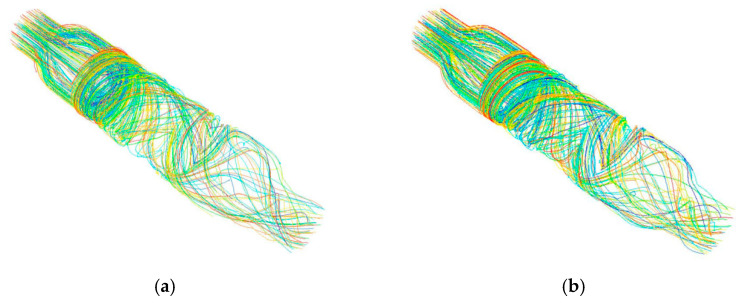
Streamline diagram of the (**a**) non-integrated design and (**b**) integrated design.

**Figure 11 sensors-22-01642-f011:**
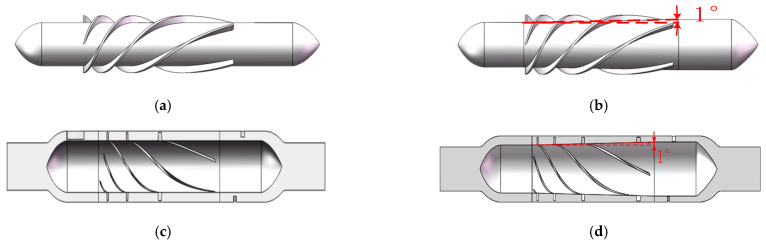
Different rotor models: (**a**) rotor of traditional axial blood pump; (**b**) rotor of conical axial blood pump with a hub taper angle of 1°; (**c**) flow channel model of traditional axial blood pump; (**d**) flow channel model of conical axial blood pump with a hub taper angle of 1°.

**Figure 12 sensors-22-01642-f012:**
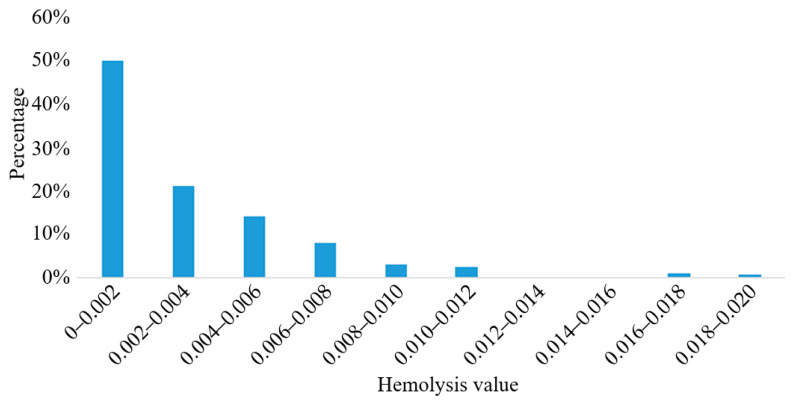
Hemolysis prediction values of blood cells.

**Figure 13 sensors-22-01642-f013:**
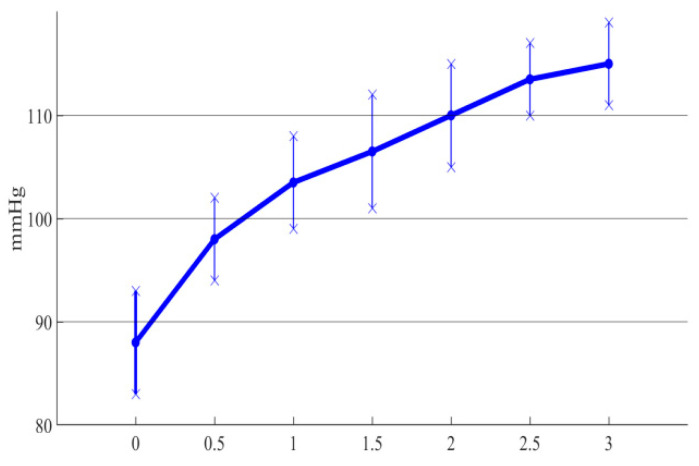
Errors between simulation and experimental results of different taper angles.

**Figure 14 sensors-22-01642-f014:**
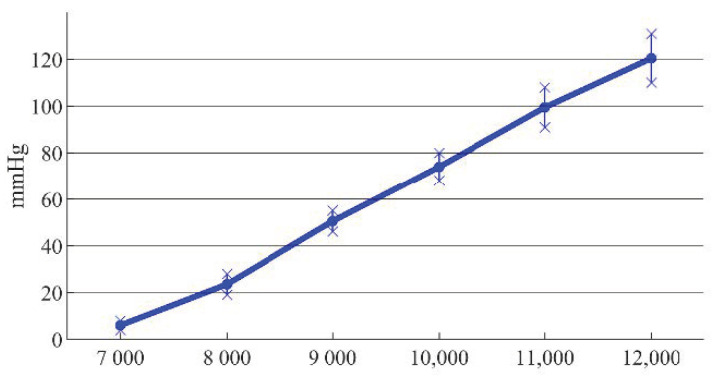
Errors between simulation and experimental results at different rotating speeds.

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
