# Peer review of "CFD-Based Flow Channel Optimization and Performance Prediction for a Conical Axial Maglev Blood Pump"

_sensors, 2022, doi:10.3390/s22041642_

Round 1

Reviewer 1 Report

The submitted paper contains original results, thus its publication can be recommended after revision.

For all details see the attached file.

Author Response

Dear Reviewers:

Thank you for your comments concerning our manuscript entitled “CFD-Based Flow Channel Optimization and Performance Pre-diction for a Conical Axial Magnetic Blood Pump” (Manuscript Number: sensors-1577970). Those comments are all very valuable for improving our paper, as well as the important guiding significance to our researches. We have studied comments carefully and have made correction which we hope meet with approval. The responses to the reviewer’s comments are in the attached file.

Reviewer 2 Report

The work was written quite clearly.
Minor comments in the attached file.

Author Response

(The authors gave the same response as above.)
